# Increase in people's behavioural risks for contracting COVID-19 during the 2021 New Year holiday season: longitudinal survey of the general population in Japan

Shuko Takahashi ,[1,2] Shohei Yamada,[1] Satoshi Sasaki,[1] Yoichi Minato,[1] Naomi Takahashi,[3] Keiichiro Kudo,[1] Masaru Nohara,[1] Ichiro Kawachi[4]

[1]Department of Health and Welfare, Iwate Prefectural Government, Morioka, Japan
[2]Division of Medical Education, Iwate Medical University, Morioka, Japan
[3]Department of Hygiene and Preventive Medicine, Graduate School of Medicine, Iwate Medical University Faculty of Medicine, Morioka, Japan
[4]Department of Social and Behavioral Sciences, Harvard University T H Chan School of Public Health, Boston, Massachusetts, USA

**Correspondence to**
Dr Shuko Takahashi;
shutakahashi-iwt@umin.ac.jp

## ABSTRACT

**Objectives** There has been no study in Japan on the predictors of risk for acquiring SARS-CoV-2 infection based on people's behaviour during the COVID-19 pandemic. The aim of this study was to document changes in risk behaviour during the New Year's holiday season in 2021 and to identify factors associated with high-risk behaviour for infection using a quantitative assessment tool.

**Design** A longitudinal survey.

**Setting** Multiphasic health check-ups for the general population in Iwate Prefecture.

**Participants** Serial cross-sectional data were obtained using rapid online surveys of residents in Iwate Prefecture from 4 to 7 December 2020 (baseline survey) and from 5 to 7 February 2021 (follow-up survey). The data in those two surveys were available for a total of 9741 participants.

**Main outcome measures** We estimated each individual's risk of acquiring SARS-CoV-2 infection based on the microCOVID calculator. We defined four trajectories of individual risk behaviours based on the probabilities of remaining at low risk, increasing to high risk, improving to low risk and persistence of high risk.

**Results** Among people in the low-risk group in the first survey, 3.6% increased to high risk, while high risk persisted in 80.0% of people who were in the high-risk group at baseline. While healthcare workers were significantly more likely to be represented in both the increasing risk and persistently high-risk group, workers in the education setting were also associated with persistence of high risk (OR 2.58, 95% CI 1.52 to 4.39; p<0.001).

**Conclusions** In determining countermeasures against COVID-19 (as well as future outbreaks), health officials should take into account population changes in behaviour during large-scale public events.

## INTRODUCTION

Cases of COVID-19 spread worldwide soon after the first case was identified in China in 2019. The worldwide death toll from COVID-19 as of 19 November 2021 is 5 139 910.[1] The first case of COVID-19 in Japan was

### Strengths and limitations of this study

► Risk behaviours associated with contracting COVID-19 were assessed in a large sample of the general population living in one Japanese prefecture.
► Longitudinal assessments were conducted at two time points, straddling the 2021 holiday season.
► The brief online survey approach allowed for rapid, real-time tracking of population changes in behaviour.
► A limitation of the study was the low response rate to the baseline survey (25%) as well as the follow-up survey (40%), although the direction of the bias is likely to be conservative.

confirmed in Kanagawa on 16 January 2020, and cases then spread from urban areas to rural areas in Japan.

In December 2020, Japan experienced a third wave which was much larger than the second wave, peaking at 8045 new cases on 8 January 2021 (compared with 1575 cases at the peak of the second wave on 31 July 2020).[2] The Japanese government declared a second state of emergency for Tokyo and its surrounding prefectures on 7 January 2021, and several measures were implemented including shortening the opening hours for bars and restaurants and encouraging telework to decrease the number of commuters by 70%.[3]

One of the reasons for the increase in the number of COVID-19 cases in Japan was the movement of people during the traditional New Year holiday. New Year's Day is the most important holiday in Japan, when many people return to their hometowns to spend time with their family, relatives or local friends. The typical number of days that people take off during the New Year's holiday

**BMJ**

is about 1 week (7 days) from 28 December to 3 January. Although there is a decrease in the movement of people because schools, businesses and workplaces are closed for several days, large crowds of people gather to pray at local shrines or temples on New Year's Day (called 'Hatsu-mode' in Japanese).[4]

The Japanese government opted to not restrict the movement of people during the period before and after New Year's Day. Some people also ignored the government's request to refrain from unnecessary outings while other people continued to take preventive measures against infection. However, the number of people who ignored government directives is not known. Studies have shown that there are variations in the tendency to adopt protective measures such as social distancing based on people's age, gender, socioeconomic characteristics (educational attainment, occupation) and political ideology.[5–8] However, there were a few tools to examine people's daily behaviour to quantify the risk of being infected based on people's daily activities. In addition, the extent of contagion differs with regional conditions, as well as population mobility during special events such as national holidays. In order to establish effective interventions during large-scale public events, there is a need to identify high-risk groups using an objective tool.

The aim of our study was to determine different groups of individuals based on their trajectories of risk behaviour during the New Year holiday season and to determine factors associated with behaviour with a high risk for COVID-19 using a quantitative risk assessment tool, 'microCOVID'.

## MATERIALS AND METHODS
### Study area
Iwate Prefecture is located in the north-eastern part of Japan (about 500 km from Tokyo) with a population of about 1.2 million (online supplemental figure S1). The total number of COVID-19 cases as of 19 November 2021 was 3487 with 53 cases of COVID-19-related deaths reported.[9]

### Data
Rapid online surveys of residents in Iwate Prefecture were conducted by the Office of Medical Policy in the Department of Health and Welfare in Iwate Prefecture Government using a popular social network platform called LINE (LINE, Tokyo, Japan). By using the LINE app, Iwate Prefecture has been providing information about COVID-19 every day since the beginning of the pandemic, including information on the number of new cases and characteristics of patients. A series of cross-sectional surveys were started in December 2020 to investigate people's behaviour in order to prevent infection and promote health consciousness. The surveys have been conducted every 2 months. An online questionnaire was administered to a total of 100 958 people who had registered by the time of the baseline survey. We sent out

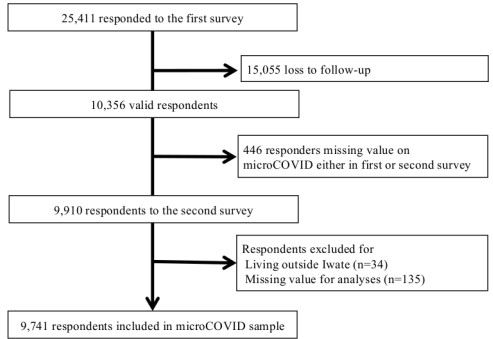

**Figure 1** Flow chart of the selection of respondents. Of 100 958 registered residents, 9741 were selected for this study.

notification about the survey via the LINE platform on 4 December 2020, and 25 411 individuals answered from 4 to 7 December (participant rate, 25.2%). We conducted a second survey of registered people from 5 to 7 February 2021. We analysed the results of these two surveys. After excluding individuals who were lost to follow-up in the second survey (15055), those who were missing data needed for the calculation of microCOVID scores either in first or second survey (n=446), those who live outside Iwate (n=34) and those who had other missing values for analyses (n=135), data were available for a total of 9741 participants (figure 1).

### Outcome
We applied a weighting system to calculate the level of behavioural risk for acquiring COVID-19 infection, called microCOVID.[10] microCOVID is a calculator to numerically quantify the risk of getting COVID-19 from daily activities. 1.0 microCOVID is equivalent to a one-in-a-million chance of getting COVID-19. microCOVIDs are computed by using three major factors: activity risk, person risk and number of people with whom an individual interacts with (online supplemental text S1). We obtained the microCOVID value for each person whereby the score=activity risk × number of people × person risk.

Activity risk indicates the chance that an activity will transmit COVID-19 from a person who currently is infected with COVID-19. According to microCOVID, the transmission risk is estimated to be 'about 9% per hour' from spending more than 10 min indoors or in close proximity with an unmasked person who is COVID-19 positive. Coefficients of risk are assigned to other types of interaction which differ from that reference value. The coefficients were calculated based on the following factors: duration of interaction, mask wearing (respondents and other persons), indoor/outdoor environment, distance from each other, volume of conversation and frequency (times a week) (online supplemental table S1). With regard to the number of people, we asked respondents 'how many people were there within a 5 meter radius of the scene?' Person risk represents the probability that a random person is currently infected with COVID-19 based on the overall prevalence in the person's area as well as the recent behaviours of the person (online

supplemental table S1). We classified each individual's risk level into low risk versus high risk for acquiring infection (low-risk group, ≤20 microCOVIDs; high-risk group, >20 microCOVIDs).

## Covariates

Age, sex, municipality, occupation, frequency of eating out compared with the past year, returning home during the New Year holiday and visits to the local shrine/temple were included as independent variables in the analysis. Participants were divided into three age categories: young (people less than 39 years of age), middle age (people of 40–59 years of age) and elderly (people aged 60 years or older). Residential areas were classified into inland versus coastal/mountainous based on the geography of Iwate Prefecture. Occupations were assessed by asking 'what is your current job?' Participants were divided into five groups: healthcare workers, workers in service industries (eg, transportation, customer-facing occupations in the retail/hospitality sector, office workers), education sector (teachers or students), government workers and all others (workers in manufacturing, farmers/agricultural workers, workers in other jobs or unemployed). The frequency of eating out compared with the past year was classified into three groups: 'decreased by 80% or more', 'decreased by 50%–70%' and 'decreased by 40% or less'. With regard to the first shrine visit of the year, participants were divided into three categories: 'never observe the ceremony', 'visited' and 'did not visit this year as a preventive measure'. We asked respondents about the extent of preventive measures adopted during the second state of emergency compared with that in the first one. The participants were divided into three groups: lower (lower degree or no preventive measure), same (same as the first one) and higher (higher degree).

## Statistical analyses

For analysis of the trajectories of behaviour risks, we calculated the risk scores for each individual in the first survey and that in the second survey. Baseline characteristics were compared for the low-risk and high-risk microCOVID groups in the second survey using the $\chi^2$ test. Logistic regression analyses were conducted to identify the characteristics related to high-risk behaviour. We conducted analyses separated by low risk and high risk in the first survey to determine the predictors of each type of trajectory. We imputed missing covariate data by multiple imputation using Markov chain Monte Carlo method, creating five imputed data sets. We used the Statistical Package for Social Sciences (SPSS) software program V.25.0 (IBM) for all analyses. All statistical tests described were two sided, and analysis items with p values <0.05 were considered statistically significant.

## Patient and public involvement

No funding was available for patient or public involvement in this project. No patients were involved in setting the research question or the outcome measures. Patients were not invited in the design, or conduct, or reporting, or dissemination plans of our research.

## RESULTS

Figure 2 shows the trajectories of behavioural risks from the first survey to the second survey. In December 2020, 17.6% of the respondents were classified into the high-risk behaviour category. Two months after the first survey (straddling the New Year holidays), 3.6% of the low-risk individuals in the initial survey had transitioned to high risk, while 80.0% of the individuals in the high-risk behaviour category in the baseline survey remained in the high-risk group. The percentage of participants in the low-risk group was higher in the second survey than in the first survey, reflecting the decline in overall prevalence in Iwate Prefecture by the time of the follow-up survey.

Table 1 shows the baseline characteristics of participants in the low-risk group and high-risk group at the baseline survey. Individuals in the low-risk group at the baseline survey who later transitioned to high risk included a higher proportion of young people compared with individuals who remained in the low-risk group. Healthcare workers were also over-represented in the group who transitioned to high risk (35.0%) compared with those who remained at low risk (16.5%, p<0.001). The high-risk group at baseline also included people in education (teachers and students), as well as individuals who reported never observing traditional visits to shrines/temples during New Year.

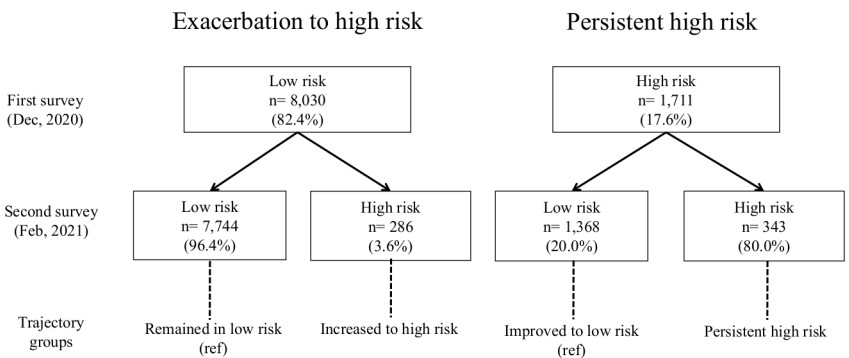

**Figure 2** Analytical models for behavioural risk trajectories.

**Table 1** Baseline characteristics of participants in the first survey (n=9741)

| | | Low risk in the first survey (n=8030) | | | High risk in the first survey (n=1711) | | |
| --- | --- | --- | --- | --- | --- | --- | --- |
| | | Remained in low risk (n=7744) | Increased to high risk (n=286) | | Improved to low risk (n=1368) | Persistent high risk (n=343) | |
| | | n (%) | n (%) | P value | n (%) | n (%) | P value |
| Age groups | Young | 2127 (27.5) | 108 (37.8) | <0.001* | 414 (30.3) | 116 (33.8) | 0.085 |
| | Middle age | 4577 (59.1) | 161 (56.3) | | 852 (62.3) | 212 (61.8) | |
| | Elderly | 1040 (13.4) | 17 (5.9) | | 102 (7.5) | 15 (4.4) | |
| Sex | Women | 5476 (70.7) | 211 (73.8) | 0.263 | 1078 (78.8) | 284 (82.8) | 0.100 |
| Area | Inland areas | 6250 (80.7) | 232 (81.1) | 0.863 | 1107 (80.9) | 270 (78.7) | 0.357 |
| Occupation | Healthcare workers | 1274 (16.5) | 100 (35.0) | <0.001* | 519 (37.9) | 146 (42.6) | 0.001* |
| | Service | 2295 (29.6) | 87 (30.4) | | 385 (28.1) | 91 (26.5) | |
| | Schools | 600 (7.7) | 24 (8.4) | | 116 (8.5) | 48 (14.0) | |
| | Others | 2671 (34.5) | 45 (15.7) | | 167 (12.2) | 30 (8.7) | |
| | Government workers | 904 (11.7) | 30 (10.5) | | 181 (13.2) | 28 (8.2) | |
| Rate of decrease in eating out compared with that in the past year | Decreased by 80% or more | 4517 (58.3) | 159 (55.6) | 0.378 | 781 (57.1) | 183 (53.4) | 0.343 |
| | Decreased by 50%–70% | 1673 (21.6) | 60 (21.0) | | 296 (21.6) | 86 (25.1) | |
| | Decreased by 40% or less | 1554 (20.1) | 67 (23.4) | | 291 (21.3) | 74 (21.6) | |
| Homecoming in the New Year's holiday season | Yes | 1189 (15.4) | 50 (17.5) | 0.328 | 231 (16.9) | 70 (20.4) | 0.126 |
| The first shrine visit of the year | Do not visit every year | 1135 (14.7) | 43 (15.0) | 0.954 | 172 (12.6) | 70 (20.4) | 0.001* |
| | Visited | 3809 (49.2) | 142 (49.7) | | 734 (53.7) | 174 (50.7) | |
| | Did not visit this year for prevention | 2800 (36.2) | 101 (35.3) | | 462 (33.8) | 99 (28.9) | |
| Measures in the second state of emergency | Lower | 741 (9.6) | 43 (15.0) | 0.007* | 155 (11.3) | 59 (17.2) | 0.003* |
| | Same | 5345 (69.0) | 180 (62.9) | | 933 (68.2) | 204 (59.5) | |
| | Higher | 1658 (21.4) | 63 (22.0) | | 280 (20.5) | 80 (23.3) | |

Categorical variables are presented as number of cases (%).
P values were calculated using the $\chi^2$ test for categorical variables.
*Statistically significant differences between two groups.

Table 2 summarises the predictors of the trajectories of behavioural risks in the two models. With regard to transitioning to high risk, younger individuals as well as individuals practising a lower degree of preventive measures during the second state of emergency had significantly higher ORs. With regard to occupation, both healthcare workers and people in educational settings were more likely to be in the group with persistent high risk (OR 2.58, 95% CI 1.52 to 4.39; p<0.001). Results were similar in the analyses with multiple imputation for missing data (n=9876) (online supplemental table S2).

## DISCUSSION

We evaluated the predictors of high-risk behaviour for SARS-CoV-2 infection during the Japanese New Year's holiday season in one prefecture. Overall, only a small percentage (<4%) of individuals transitioned from low risk (before the holidays) to high risk. However, the majority of individuals (80%) who were in the high-risk behaviour group at baseline remained high risk at follow-up.

The characteristics of individuals transitioning to high-risk behaviour included younger age, people working in the healthcare sector, as well as individuals who admitted to practising a lower degree of preventive measures during the state of emergency. Risk factors for persistently

**Table 2** Results of analysis using models for risk trajectories (n=9741)

| | | Model for transition to high risk (n=8030) | | Model for high-risk persistence (n=1711) | |
|---|---|---|---|---|---|
| | | OR (95% CI) | P value | OR (95% CI) | P value |
| Age groups | Young (ref: Elderly) | 2.22 (1.29 to 3.79) | 0.004 | 1.54 (0.84 to 2.80) | 0.16 |
| | Middle age | 1.60 (0.96 to 2.69) | 0.073 | 1.50 (0.84 to 2.68) | 0.166 |
| Sex | Women (ref: Men) | 1.01 (0.76 to 1.33) | 0.952 | 1.26 (0.91 to 1.74) | 0.169 |
| Area | Inland areas (ref: Coastal and mountainous areas) | 1.07 (0.79 to 1.46) | 0.662 | 0.86 (0.63 to 1.16) | 0.309 |
| Occupation | Healthcare workers (ref: Government workers) | 2.49 (1.62 to 3.82) | <0.001 | 1.85 (1.18 to 2.90) | 0.007 |
| | Service | 1.16 (0.76 to 1.78) | 0.502 | 1.55 (0.97 to 2.47) | 0.067 |
| | Schools | 1.11 (0.64 to 1.93) | 0.709 | 2.58 (1.52 to 4.39) | <0.001 |
| | Others | 0.55 (0.34 to 0.89) | 0.016 | 1.20 (0.68 to 2.12) | 0.525 |
| Rate of decrease in eating out compared with that in the past year | Decreased by 50%–70% (ref: Decreased by 80% or more) | 0.98 (0.72 to 1.34) | 0.918 | 1.24 (0.92 to 1.68) | 0.158 |
| | Decreased by 40% or less | 1.25 (0.92 to 1.68) | 0.151 | 1.04 (0.76 to 1.43) | 0.8 |
| Homecoming in the New Year's holiday season | Yes (ref: No) | 1.06 (0.77 to 1.46) | 0.736 | 1.18 (0.86 to 1.61) | 0.299 |
| The first shrine visit of the year | Do not visit every year (ref: Not visited on this year for prevention) | 1.05 (0.73 to 1.51) | 0.805 | 1.83 (1.28 to 2.61) | 0.001 |
| | Visited | 1.02 (0.79 to 1.33) | 0.871 | 1.09 (0.83 to 1.44) | 0.536 |
| Measures in the second state of emergency | Lower (ref: Higher) | 1.62 (1.08 to 2.43) | 0.021 | 1.28 (0.85 to 1.91) | 0.233 |
| | Same | 0.92 (0.69 to 1.24) | 0.592 | 0.75 (0.56 to 1.01) | 0.056 |

CI, confidence interval; OR, odds ratio.

high risk were mainly based on occupation, viz working in the healthcare sector and education sector.

Previous studies have examined factors associated with risk behaviours during COVID-19.[11–13] Factors related to preventive knowledge and behaviour for COVID-19 have been examined in several studies.[5–8] Shahnazi *et al* examined the associations between preventive behaviours for COVID-19 and health belief factors, fatalistic beliefs and demographic factors among 750 Iranian adults during the period from 11 to 16 March 2020. They showed that the adoption of preventive behaviour was higher in women than men and higher in urban dwellers than rural dwellers.[5] Chen and Chen examined the differences between preventive behaviours for COVID-19 of urban and rural residents in China (n=1591) from 31 January to 4 February 2020.[6] Their results showed that rural residents were less likely to practise preventive behaviours. Li *et al* examined the associations of internet use, risk awareness and demographic characteristics with engagement in preventive behaviours in the USA (n=979) from 10 to 14 April 2020.[7] They showed that women and older participants were more likely to adopt preventive behaviour, and conversely more educated participants were less likely to adopt preventive behaviour. An *et al* investigated the characteristics related to positive and negative attitudes towards social distancing as a preventive measure in the

USA (n=1074) in May 2020.[8] They showed that female sex, older ages (65 years or older) and Democratic political party support were associated with a more positive attitude towards social distancing. Our study adds to the literature by tracking changes in population behaviour during a national public holiday.

Identifying high-risk occupational settings for acquiring COVID-19 infection remains an important goal for policy, for example, prioritising vaccinations. Healthcare work was recognised as a high-risk occupation during the 2009 influenza A (H1N1) pandemic.[14] Occupational risk in the current COVID-19 pandemic has been examined in several studies.[15–17] Nguyen *et al* showed that the hazard ratio of the risk for COVID-19 was 3.40 (95% CI 3.37 to 3.43) among front-line healthcare workers.[16] Mutambudzi *et al* reported that the risk ratio for testing positive for COVID-19 or death caused by COVID-19 was 7.43 (95% CI 5.52 to 10.0) among healthcare workers.[17] These results are consistent with our results showing that healthcare workers were over-represented in both the persistently high-risk group as well as the group transitioning from low to high risk. Although we could not conduct a study to validate the assessment tool due to an insufficient number of patients with COVID-19 (n=15) among the respondents, an ecological survey by occupation that was conducted after the baseline survey showed

that the proportion of COVID-19 infections among healthcare workers was high in Iwate Prefecture.

Interestingly, people in the education sector had substantially higher ORs for high-risk persistence even though they did not gather in schools for 3 weeks during the winter vacation. After our investigation period, which included 2 weeks after the second survey during the period from 8 December 2020 to 21 February 2021, about 14% of the total number of patients with COVID-19 was reported to be among teachers and school-age children in Iwate. A meta-analysis showed that children and adolescents have lower susceptibility to SARS-CoV-2; the pooled OR of being an infected contact in children compared with that in adults was 0.56 (95% CI 0.37 to 0.85).[18] A study on contact tracing showed a lower level of transmission from index cases of children or teachers.[19] In those studies, the susceptibility to and transmission of SARS-CoV-2 in children or adolescents were compared with those in adults, and there was no comparison of the risks for teachers and students with the risks for other workers. Nguyen *et al* also reported that social and education workers had a significantly higher risk for COVID-19 than did non-essential workers in the UK and the USA.[16] The government in Japan closed the schools at the start of the first wave of COVID-19 but did not request the schools to close in the second and third waves, possibly based on the prediction of a low rate of infections in schools and less severe cases of COVID-19 in children.[20] Although fewer cases of infection and death have been reported in children and adolescents,[21] a study conducted in Sweden showed that parents, teachers and partners exposed to open schools had significantly higher ORs than those in closed schools.[22]

Initially, we hypothesised that people who made visits to shrines during the holidays (ie, mingling with crowds) would have high risk for SARS-CoV-2 infection. In fact, our study did not suggest increased risk, possibly because of near universal mask wearing, and the fact that praying at shrines and temples occurs outdoors. On the other hand, although there are some other risky behaviours that typically occur during the New Year holiday season in Japan, we could not collect detailed information on these events, for example, 'Bonen-kai' (end-of-year parties in workplaces),[23] and large indoor gatherings of extended families during the New Year.

### Limitations

The present study had several limitations. First, we did not ask questions correlated with healthy behaviours such as educational attainment or household income. We also did not ask questions about job titles, which might influence the risk of exposure to potentially high-risk environments. Second, there may have been selection bias since the respondents were limited to people registered in a health information programme about COVID-19 maintained by the Iwate Prefecture Government. We compared the characteristics of the participants at baseline (December 2020) with the characteristics of the whole population in Iwate in 2020 (online supplemental table S3). While the percentages of young adults and elderly were smaller in our study than in the census data, the percentages of middle-aged people and women were higher in our study than in the census data (online supplemental table S3). There was a large percentage of older women in the participants in our study. We reran the analyses using survey weights based on demographic characteristics obtained from the census, including sex, age group and residential area (online supplemental table S4). In weighted analyses, younger individuals, healthcare workers and people living in inland areas continued to show significantly higher ORs for transitioning to high risk. On the other hand, the ORs for persistently high risk became statistically non-significant for healthcare workers and people in educational settings. Third, only 40% of the participants in the initial survey participated in the second survey. We compared the baseline characteristics of the non-participants (n=15 055) and participants (n=10 356) at the second survey (online supplemental table S5). The percentage of younger men, people living in the inland area as well as healthcare workers was higher in non-participants compared with participants. Based on previous studies, people who respond to surveys about COVID-19 tend to have a higher level of consciousness about avoiding exposure/infection compared with non-respondents. Thus, we believe the direction of our attrition bias is conservative, that is, we may have underestimated the increase in high-risk behaviour. Despite these limitations, the important strengths of our study were the large sample of participants who were followed at two different time points and the determination of predictors of risky behaviour for COVID-19 during the New Year holiday season. This enabled us to determine the trajectories of high-risk behaviours straddling a large-scale public event.

### CONCLUSION

Healthcare workers and people in the education sector (teachers and pupils) were found to have the highest behavioural risk during the New Year holiday season in Japan. Our results offer some clues for who should be prioritised for COVID-19 vaccination as the programme is rolled out in Japan.

**Acknowledgements** The authors would like to thank the members in the Office of Medical Policy in the Department of Health and Welfare as well as the healthcare workers who contributed to active epidemiological investigation.

**Contributors** All authors have participated in the concept and design, interpretation of data, critical revision and drafting or revising of the manuscript. ST: funding acquisition, conceptualisation, formal analysis and writing–original draft. SY: verified the underlying data, data curation, resources and writing–review and editing. SS and YM: investigation, project administration and writing–review and editing. NT: investigation and writing–review and editing. KK and NM: project administration, resources, supervision and writing–review and editing. IK: methodology, supervision, validation and writing–review and editing. ST acts as guarantor for the manuscript.

**Funding** The work was supported by JSPS KAKENHI (grant number JP20K18858).

**Disclaimer** The funders had no role in the study design, data collection and analysis, decision to publish or preparation of the manuscript.

**Map disclaimer** The inclusion of any map (including the depiction of any boundaries therein), or of any geographic or locational reference, does not imply the expression of any opinion whatsoever on the part of BMJ concerning the legal status of any country, territory, jurisdiction or area or of its authorities. Any such expression remains solely that of the relevant source and is not endorsed by BMJ. Maps are provided without any warranty of any kind, either express or implied.

**Competing interests** None declared.

**Patient consent for publication** Not applicable.

**Ethics approval** The surveys were conducted in accordance with the applicable Japanese law and the Iwate prefectural government policy. Since this study was a secondary analysis using anonymised data from Iwate Prefecture, the study was exempt from Institutional Review Board review.

**Provenance and peer review** Not commissioned; externally peer reviewed.

**Data availability statement** Data are available upon reasonable request. The data that support the findings of this study are available from the corresponding author (ST) and the Division of Health and National Health Insurance in the Department of Health and Welfare, Iwate Prefecture Government, but restrictions apply to the availability of these data, which were used under licence for the current study, and so are not publicly available. Data are however available from the authors upon reasonable request and with permission of the Division of National Health Insurance in the Department of Health and Welfare, Iwate Prefecture Government.

**ORCID iD**
Shuko Takahashi http://orcid.org/0000-0002-7384-1113

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
