## [Reviewer comments · BMJ Open]

ARTICLE DETAILS

TITLE (PROVISIONAL)	Increase in people's behavioral risks for contracting COVID-19 during the 2021 New Year holiday season: Longitudinal survey of the general population in Japan
AUTHORS	Takahashi, Shuko; Yamada, Shohei; Sasaki, Satoshi; Minato, Yoichi; Takahashi, Naomi; Kudo, Keiichiro; Masaru, Nohara; Kawachi, Ichiro

VERSION 1 – REVIEW

REVIEWER	Chan, Ho Fai Queensland University of Technology
REVIEW RETURNED	31-Jul-2021

GENERAL COMMENTS	This study attempts to study the factors associate with increase in COVID-19 exposure risk during the New Year holiday season in the Iwate Prefecture, Japan. Using the microCOVID assessment tool implemented in two survey waves, the authors have identified that healthcare workers and the younger population are more likely to increase risk exposure. Those in the education sector were also found to remain in the high risk category. The objectives of the paper are clear and is generally well-written, however, I have some concerns regarding the method and analysis and the interpretation of the results which I listed below. Please address all concerns raised. 1. As the authors stated in the limitation section, representativeness of the sample is a major concern of the study, especially when the overall participation rate is low (~25%). A remedy of this would be apply survey weights when analyzing the data.2. Furthermore, the concern over the sample selection effect is also fueled by the low retention rates in the second survey (only ~40% of those participated in the first survey entered the second survey). While the sample characteristics of the sample included in the analysis were compared to the population in Table S3, I would like to see the distribution of the sample characteristics of those who did not participated in the second survey. This will give more insights into the possible sample selection bias of the current result. Especially, the analysis of change in risk exposure would be biased if attrition is correlated with the covariates included in the logistic analysis as well as other unobservable characteristics.3. Were all participants over 18 years old (legal age)? I think the range of the 'young' age group is too large, the behavior of young adults (18-25 or 18-30) might be different to those that are older. Please check the robustness of the results using alternative age group (e.g., 18-30, 31-45, 45-60, 60+).4. Occupation: what is the share of the sample who is
---

	unemployed? Are they classified under 'Other'? Do occupation change between the two surveys (e.g., became unemployed)? 5. The microCOVID measure is quite refine, however, I don't think the current study is taking its full advantage; for example, why not use microCOVID directly as the outcome variable when analyzing change in risk exposure? Or employ more levels (very low, low, moderate, high, very high) risk? 6. The shrine visit variable is interesting, however, why do the author decide not to use it as an outcome variable? Particularly when response record the behavioural choice of "did not visit this year as a preventive measure". Similarly, one could also use the variable "preventive measures adopted during the second state of emergency compared with that in the first one" and "frequency of eating out" as an outcome variable in measuring behavioral change. Nevertheless, the former variable needs further clarification of how it is measured. 7. Overall, the share of participants in the low risk group is larger in the second survey, compared to the first survey (13.5% increase in low risk group), is this reflective of the COVID-19 situation in the Iwate Prefecture in general? I think this point is worth pointing out. 8. In general, I think the study missed some important literature on risk-taking behaviour/risk attitude during COVID-19 concerning mobility. Minor comments: 9. What is the explained variation of the models presented in Table 2? 10. There is a misprint in Table 1 for the number of observations for "Decreased by 80% or more" of "Persistent high risk".
--	---

REVIEWER	Marin, Benjamin Brown University
REVIEW RETURNED	30-Sep-2021

GENERAL COMMENTS	Thank you for your manuscript. Below, I have included some suggestions to further strengthen your work: Line 64 - Could you please update these figures to reflect more recent ones? Line 65 - Could you include the city in which the first case of COVID-19 was detected in Japan? Line 68 - Please tell us the number of cases seen in the second wave to help the reader better understand how it was different from the third wave. Line 74 - I appreciate the context of the New Year's Day holiday in Japan. Could you clarify the rough number of days that individuals attend to this holiday? (is it only one day, or two days, or is a whole week "given off" by employers?) This should help the reader better understand the timeframe of these festivities. This will in turn clarify your statement in Lines 80-81. Line 81 - Are there estimates available of how many people ignored these government suggestions? If not, please state so Line 91 - Could you please explain here what "microCOVID" is and why it is the most appropriate assessment tool for your study? What
---

	about it makes it the best tool to answer your research question? You currently discuss this in the Outcome section of the paper, but I recommend answering the above aspects of microCOVID in the introduction. You may then choose to keep the more technical aspects of microCOVID where they currently are. Also, please provide context of how microCOVID has been used by others / what kind of research questions it has already answered. Line 97 - Please consider updating numbers Line 102 - Please provide an estimate of how many LINE users exist in Japan and/or in Iwate Prefecture Line 107 - Could you justify why the surveys were conducted every two months. Line 172 - can you briefly justify why Markov Chain Monte Carlo is the most appropriate approach for your analysis? Line 193 - Can you clarify the extent of health care worker over-representation this group? Discussion - Could you offer a few recommendations to reduce Covid spread during the next New Year season in Japan, based on your findings?
--	--

VERSION 1 – AUTHOR RESPONSE

Reviewer: 1

This study attempts to study the factors associate with increase in COVID-19 exposure risk during the New Year holiday season in the Iwate Prefecture, Japan. Using the microCOVID assessment tool implemented in two survey waves, the authors have identified that healthcare workers and the younger population are more likely to increase risk exposure. Those in the education sector were also found to remain in the high risk category. The objectives of the paper are clear and is generally well-written, however, I have some concerns regarding the method and analysis and the interpretation of the results which I listed below. Please address all concerns raised.

Major revision

1) As the authors stated in the limitation section, representativeness of the sample is a major concern of the study, especially when the overall participation rate is low (~25%). A remedy of this would be apply survey weights when analyzing the data.

Our response:

Firstly, we have corrected the number of loss to follow-up from 14,998 to 15,055. We have revised the figure in the Figure 1.

There was a larger percentage of older women among our survey respondents compared with the local Census data of Iwate prefecture residents (S3 Table). We repeated our analyses using survey weights calculated from sex, age groups, and residential area (R1 Table 1). In weighted analyses, younger individuals, health care workers and people living in inland areas continued to show significantly higher odds ratios (ORs) of transitioning to higher risk behaviors during the holiday period. With regard to persistence of high risk behavior, while both health care workers and people in educational settings tended to have higher ORs, the estimated were no longer statistically significant. We believe that the main findings of our study are robust; in particular, health care workers increased their behavioral risks over the New Year holiday season.

We have summarized these analyses in S4 Table, and have added some comments in the Limitation section.

(page 18, line 305 to 310)

“We re-ran the analyses using survey weights based on demographic characteristics obtained from the Census, including sex, age group, and residential area (S4 Table). In weighted analyses, younger individuals, health care workers, and people living in inland areas continued to show significantly higher ORs for transitioning to high risk. On the other hand the ORs for persistently high risk became statistically non-significant for health care workers and people in educational settings.”

S4 Table (R1 Table 1). ABOUT HERE

2) Furthermore, the concern over the sample selection effect is also fueled by the low retention rates in the second survey (only ~40% of those participated in the first survey entered the second survey). While the sample characteristics of the sample included in the analysis were compared to the population in Table S3, I would like to see the distribution of the sample characteristics of those who did not participate in the second survey. This will give more insights into the possible sample selection bias of the current result. Especially, the analysis of change in risk exposure would be biased if attrition is correlated with the covariates included in the logistic analysis as well as other unobservable characteristics.

Our response:

We provided a comparison of baseline characteristics of the non-participants (n= 15,055) and participants (n=10,356) at the follow-up survey (R1 Table 2). The percentage of younger men, people living in the inland area as well as health care workers was higher in non-participants compared to participants. Based on previous studies, people who respond to surveys about COVID-19 tend to have a higher level of consciousness about avoiding exposure/infection compared to non-respondents. Thus, we believe the direction of our attrition bias is conservative, i.e., we may have under-estimated the increase in high risk behavior.

We have added R1 Table 2 as S5 Table and have added the following text in the Limitations section in accordance with the reviewer’s comments.

(Page 18, line 310 to page 19, line 318).

“Third, only 40% of the participants in the initial survey participated in the second survey. We compared baseline characteristics of the non-participants (n= 15,055) and participants (n=10,356) at the second survey (S5 Table). The percentage of younger men, people living in the inland area as well as health care workers was higher in non-participants compared to participants. Based on previous studies, people who respond to surveys about COVID-19 tend to have a higher level of consciousness about avoiding exposure/infection compared to non-respondents. Thus, we believe the direction of our attrition bias is conservative, i.e., we may have under-estimated the increase in high risk behavior.”

S3 Table. ABOUT HERE

S5 Table (R1 Table 2). ABOUT HERE

3) Were all participants over 18 years old (legal age)? I think the range of the 'young' age group is too large, the behavior of young adults (18-25 or 18-30) might be different to those that are older. Please check the robustness of the results using alternative age group (e.g., 18-30, 31-45, 45-60, 60+).

Our response:

No, subjects in the present survey was not over the age of 18. Because we did not ask participants their age, we included subjects who are aged 0-17 years. In the present study, participants selected one group from seven age-groups: "under 20 years", "aged 20-29 years", "aged 30-39 years", "aged 40-49 years", "aged 50-59 years", "aged 60-69 years", and "age 70 years old or older".

Following the reviewer's suggestion, we classified age groups into four groups; "under 30 years", "30-49 years", "50-59 years", and "60 years or older". We can break down "under 30 years", but the sample size in people aged under 20 years is too small (n=126, 1.3 % of the analytic sample).

Therefore, we combined people aged under 20 years and those aged 20-29 years into one group "under 30 years".

The number of participants (%) were 2,583 (7.7%) in people under 30 years, 13,297 (39.4%) in those aged 30-49 years, 6405 (19.0%) in those aged 50-59 years, and 3,126 (9.3%) in those aged 60 years or older.

R1 Table 3 summarizes the predictors of the trajectories of behavioral risks in the two models. The participants aged under 30 years and those aged 30-49 years had significantly higher ORs than those aged 60 years or older, but the magnitude of ORs did not differ between people under 30 years versus those aged 30-49 years.

R1 Table 3. ABOUT HERE

4) Occupation: what is the share of the sample who is unemployed? Are they classified under 'Other'? Do occupation change between the two surveys (e.g., became unemployed)?

Our response:

The people who answered unemployed were classified into the "others" group. R1 Table 4 shows the change in the prevalence of unemployment between the first survey and the second survey. There was no significant change in unemployment between the two surveys (P= 0.882).

R1 Table 4. ABOUT HERE

5) The microCOVID measure is quite refine, however, I don't think the current study is taking its full advantage; for example, why not use microCOVID directly as the outcome variable when analyzing change in risk exposure? Or employ more levels (very low, low, moderate, high, very high) risk?

Our response:

In accordance with the reviewer's suggestion, we calculated the ORs for quartiles of microCOVID (very low, low, moderate, and high) using multinomial logistic regression analysis (R1 Table 5). With regard to transitioning to higher risks among the very low-risk and low-risk groups in the first survey, younger individuals showed significantly higher ORs. With regard to occupation, while health care workers in the high-risk group at the first survey tended to remain in the high-risk group, health care workers initially in the very low-risk/low-risk, or moderate-risk groups at the first survey were significantly more likely to transition to the high risk group at the second survey (OR [95% confidence interval (CI)]: very low-risk group, 1.59 [1.06 - 2.38]; low-risk

group, 2.69 [1.39 - 5.19]; moderate-risk group, 1.58 [1.04 - 2.39]). People working in educational settings were more likely to remain in the persistently high risk group (OR [95% CI], 1.93 [1.18 - 3.13]; $P < 0.008$). In summary, these results were similar to our initial analyses where we classified microCOVID into two groups: low vs. high.

There are two reasons why we presented our results using a dichotomized classification of microCOVID. First, we were following the original cut-points suggested by the microCOVID project 1. Second, since our results from the more refined categorization of microCOVID did not differ from our original approach, we have decided to keep our original approach for reasons of simplicity in presentation.

R1 Table 5. ABOUT HERE

6) The shrine visit variable is interesting, however, why do the author decide not to use it as an outcome variable? Particularly when response record the behavioural choice of “did not visit this year as a preventive measure”.

Similarly, one could also use the variable “preventive measures adopted during the second state of emergency compared with that in the first one” and “frequency of eating out” as an outcome variable in measuring behavioral change. Nevertheless, the former variable needs further clarification of how it is measured.

Our response:

The first shrine visit of the year is a traditional custom in Japanese Shinto religion. It was assessed by asking the question “Please tell us whether you visited a shrine”. Participants were asked to choose one of four responses: “did not visit this year to avoid getting COVID”, “do not visit every year”, “visited without any concerns”, and “visited by shifting the days and times”. Based on their answers, participants were categorized into three groups: not visited on this year to avoid infection, do not visit every year, and visited (combined categories of visited without any concerns and visited by shifting the days and times).

Following the reviewer’s suggestion, we used shrine visit as an outcome variable. R1 Table 6 (a) shows the predictors of shrine visits. Although people living in inland areas had significantly higher ORs of not visiting this year to avoid infection (compared with people living in coastal & mountainous areas), we did not find other associations that we could easily interpret. We also conducted similar analyses using “preventive measures adopted during the second state of emergency compared with that in the first one” and “frequency of eating out” as outcome variables. With regard to predictors of “decrease in eating out compared with the past year”, people who visited the local shrine showed significantly higher ORs (R1 Table 6 (b)). With regard to predictors of preventive measures adopted during the second state of emergency, young and middle-aged people showed significantly higher ORs of engaging in fewer preventive measures during the second state of emergency compared with the elderly (R1 Table 6 (c)).

R1 Table 6 (a) (b) (c) ABOUT HERE

7) Overall, the share of participants in the low risk group is larger in the second survey, compared to the first survey (13.5% increase in low risk group), is this reflective of the COVID-19 situation in the Iwate Prefecture in general? I think this point is worth pointing out.

Our response:

The reviewer's point is correct. microCOVID values were calculated by considering the overall prevalence in the person's area. Since the overall prevalence in Iwate was lower in the second survey than in the first survey, the percentage of participants in the low-risk group was larger in the second survey.

In accordance with the reviewer's point, we have added this point in the Results section.

(page 9, line 195 to 197)

"The percentage of participants in the low-risk group was higher in the second survey than in the first survey, reflecting the decline in overall prevalence in Iwate Prefecture by the time of the followup survey."

8) In general, I think the study missed some important literature on risk-taking behaviour/risk attitude during COVID-19 concerning mobility.

Our response:

We have added the following articles in our article about risk-taking behavior/risk attitude during COVID-19 concerning mobility 2-4.

Bavel JJV, Baicker K, Boggio PS, et al. Using social and behavioural science to support COVID-19 pandemic response. *Nat Hum Behav* 2020;4(5):460-71.

Chan HF, Skali A, Savage DA, et al. Risk attitudes and human mobility during the COVID-19 pandemic. *Scientific Reports* 2020;10(1):19931.

Pan Y, Darzi A, Kabiri A, et al. Quantifying human mobility behaviour changes during the COVID-19 outbreak in the United States. *Sci Rep* 2020;10(1):20742.

We have added the text with these articles in the Discussion section.

(page 15, line 233)

"Previous studies have examined factors associated with risk behaviors during COVID-19 2-4."

Minor comments

9) What is the explained variation of the models presented in Table 2?

Our response:

We shows the R-square of microCOVID in Table 2. Nagelkerke R-Square in the low risk in the first survey was 0.050, and that in the high risk in the first survey was 0.047.

We have added those values in the Table 2.

Table 2. ABOUT HERE

10) There is a misprint in Table 1 for the number of observations for "Decreased by 80% or more" of "Persistent high risk".

Our response:

We have corrected the error.

Reviewer: 2

1) Line 64 - Could you please update these figures to reflect more recent ones?

Our response:

Thank you for reviewing our manuscript. We have updated the number of cases.

(page 4, line 64)

“The worldwide death toll from COVID-19 as of November 19, 2021 is 5,139,910 5.”

2) Line 65 - Could you include the city in which the first case of COVID-19 was detected in Japan?

Our response:

The city in which the first case of COVID-19 was detected was Kanagawa in Japan (linked to the cruise ship). We have added the name of the city in the manuscript.

(page 4, line 65)

“The first case of COVID-19 in Japan was confirmed in Kanagawa on January 16, “

3) Line 68 - Please tell us the number of cases seen in the second wave to help the reader better understand how it was different from the third wave.

Our response:

The number of new cases of COVID-19 peaked at 1,575 on July 31, 2021 during the second wave. We have added a sentence in our manuscript.

(page 4, line 67 to 69)

“In December 2020, Japan experienced a third wave which was much larger than the second wave , peaking at 8,045 new cases on January 8, 2021 (compared to 1,575 cases at the peak of the second wave on July 31, 2020) 6.”

4) Line 74 - I appreciate the context of the New Year's Day holiday in Japan. Could you clarify the rough number of days that individuals attend to this holiday? (is it only one day, or two days, or is a whole week "given off" by employers?) This should help the reader better understand the timeframe of these festivities. This will in turn clarify your statement in Lines 80-81.

Our response:

The informal number of holidays in Japan during the New Year's festivities is one week (seven days) from December 28 to January 3. People usually spent their time with their family or relatives, often returning to their hometowns.

In accordance with the reviewer's suggestion, we have added information regarding the New Year's holiday in Japan.

(page 4, line 76 to 77)

"New Year's Day is the most important holiday in Japan, when many people return to their hometowns to spend time with their family, relatives or local friends. The typical number of days that people take off during the New Year's holiday is about one week (seven days) from December 28 to January 3."

5) Line 81 - Are there estimates available of how many people ignored these government suggestions? If not, please state so

Our response:

We don't have information about the estimated number of people who ignored a request from the Government of Japan. We have added a comment in accordance with the reviewer's suggestion.

(page 4, line 82 to 85)

"Some people also ignored the government's request to refrain from unnecessary outings, while other people continued to take preventive measures against infection. However, the number of people who ignored government directives is not known."

6) Line 91 - Could you please explain here what "microCOVID" is and why it is the most appropriate assessment tool for your study? What about it makes it the best tool to answer your research question? You currently discuss this in the Outcome section of the paper, but I recommend answering the above aspects of microCOVID in the introduction. You may then choose to keep the more technical aspects of microCOVID where they currently are. Also, please provide context of how microCOVID has been used by others / what kind of research questions it has already answered.

Our response:

- microCOVID

"microCOVID" has been proposed by the microCOVID project to understand how COVID is transmitted. microCOVID is an individual-level calculator to numerically quantify the risk of being infected by SARS-Cov-2 virus from daily activities. A score of 1.0 microCOVID is equivalent to a one-in-a-million chance of being infected. microCOVIDs are computed by using three major factors: activity risk, person risk, and number of people with whom an individual interacts. We obtained the microCOVID values for each person from the score = activity risk x number of people x person risk. Please see more details on how we calculated components of microCOVID in Supplementary Text 1.

- What about it makes it the best tool to answer your research question?

Previous studies have assessed people's daily behaviors via simple questions 7-10, for example, "Whether they wear a mask when going out". However, there are more precise tools to examine people's daily behavior to numerically quantify the risk of getting COVID-19 from daily activities. We sought to estimate people's behavioral risks by evaluating people's daily behaviors across the New Year holiday season with a more objective tool. On the other hand, the extent of contagion according to the regional context. As well, the movement of people changes according major events such as the

Christmas holidays in the United States. We need to estimate the risks by considering both current epidemiological data and the timing of people's movements.

The strength of microCOVID is that it calculates people's daily behavior by considering not only the risk situation that they had experienced in the past week (activity risk) but also overall prevalence in the person's area as well as recent behaviors of the person. The purpose of our study was to therefore determine factors associated with risk behavior for becoming infected; therefore, applying a precise tool is crucial for estimating people's behavioral risks in the present study. Because microCOVID can estimate behavioral risks in real time, we believe microCOVID is the best tool to examine our research question.

- Technical aspects of microCOVID

The microCOVID project developed their calculation based on current evidence. They provide transmission risks for detailed scenarios, e.g., spending time with 2 other people outdoors while masked.

- Other researches that used microCOVID

The Behavioral Insights Team in the UK analyzed a (pre-lockdown) survey about the social activities of 3,702 adults to understand how people could be expected to comply with government guidance, in terms of limiting their number of risky social contacts after the lockdown was announced in England 11. They measured coronavirus transmission risk using microCOVID. They showed that a small minority (8%) ("potential superspreaders") could account for 60% of the total transmission risk.

In accordance with reviewer's suggestion, we have added descriptions of microCOVID in the Introduction section.

(page 5, line 87 to 92)

"However, there were a few tools to examine people's daily behavior to quantify the risk of being infected based on people's daily activities. In addition, the extent of contagion differs with regional conditions, as well as population mobility during special events such as national holidays. In order to establish effective interventions during large-scale public events, there is a need to identify high-risk groups using an objective tool."

7) Line 97 - Please consider updating numbers

Our response:

The total number of COVID-19 cases as of November 19, 2021 was 3487 with 53 cases of COVID-19-related deaths reported. We have updated it in our manuscript.

(page 5, line 101 to 103)

"The total number of COVID-19 cases as of November 19, 2021 was 3487 with 53 cases of COVID-19-related deaths reported."

8) Line 102 - Please provide an estimate of how many LINE users exist in Japan and/or in Iwate Prefecture

Our response:

The official data from LINE Corporation shows that the number of LINE users in Japan is about 89 million (70% of the total population in Japan). Although we put a question to the LINE Corporation about the number of LINE users in Iwate Prefecture, we could not get an answer.

9) Line 107 - Could you justify why the surveys were conducted every two months.

Our response:

There are two reasons why we have conducted our surveys every two months. First, the Japanese government issued new guidelines at roughly two month intervals during the first year of the pandemic, e.g., the government declared its first state of emergency due to the coronavirus on April 7, 2021 and lifted the emergency declaration on May 25, 2021. Second, we sought to minimize the burden on survey respondents by leaving a certain amount of time in between surveys.

10) Line 172 - can you briefly justify why Markov Chain Monte Carlo is the most appropriate approach for your analysis?

Our response:

Different methods are available for multiple imputation such as the monotone methods, Markov Chain Monte Carlo in the fully conditional specification, and Expectation-Maximization with Bootstrapping. Imputed variables in our dataset included five variables among eight variables and the number of missing cases was 135 (1.4%) among 9,741 participants. We did not apply the monotone method because our data did show a monotone pattern of missing values. According to M. Takahashi et. al., who compared different methods using datasets in multiple imputation, there was generally no decisive difference between the methods in terms of accuracy. Therefore, we selected the Markov Chain Monte Carlo in multiple imputation for missing values in our sensitivity analyses based on the default setting in SPSS.

11) Line 193 - Can you clarify the extent of health care worker over-representation this group?

Our response:

In the low risk in the first survey, the percentage of health care workers was significantly higher in the group transitioning to high-risk compared to those who remained at low risk in the second survey (35.0% vs. 16.5%, respectively, $P < 0.001$).

We have added this information in our manuscript.

(page 9, line 210 to line 203)

“Health care workers were also over-represented in the group who transitioned to high risk (35.0%) compared to those who remained at low risk (16.5%, $P < 0.001$).”

12) Discussion - Could you offer a few recommendations to reduce Covid spread during the next New Year season in Japan, based on your findings?

Our response:

We have some recommendations for measures to prevent COVID-19 during the upcoming New Year holidays in Japan.

1. Health officials need to take into account real time changes in population behavior during large-scale public events.
2. Health care workers and people in the education sector (teachers and pupils) should be prioritized for COVID-19 vaccination (and booster) shots, based on their high risk.
3. Policy makers need to provide timely messaging to remind the population to keep up their basic preventive measures across the New Year holiday season.

We have described these messages in the Discussion section.

References

1. Catherine OJ, Oreman Rhys, Lindmark , Anna TS, Bachar Sarah, Dobro , Matt B. microCOVID Project San Francisco, CA, USA; 2020 [updated January 2nd, 2021. Available from: https://www.microcovid.org/?distance=sixFt&duration=30&interaction=oneTime&personCount=15&riskProfile=average&setting=indoor&theirMask=none&topLocation=US_36&voice=normal&yourMask=none (accessed 24 January 2021).
2. Bavel JJV, Baicker K, Boggio PS, et al. Using social and behavioural science to support COVID-19 pandemic response. *Nat Hum Behav* 2020;4(5):460-71. doi: 10.1038/s41562-020-0884-z.
3. Chan HF, Skali A, Savage DA, et al. Risk attitudes and human mobility during the COVID-19 pandemic. *Scientific reports* 2020;10(1):19931. doi: 10.1038/s41598-020-76763-2
4. Pan Y, Darzi A, Kabiri A, et al. Quantifying human mobility behaviour changes during the COVID-19 outbreak in the United States. *Sci Rep* 2020;10(1):20742. doi: 10.1038/s41598-020-77751-2
5. Johns Hopkins University and Medicine, USA. COVID-19 Data in Motion: Monday, March 29, 2021. 2021. <https://coronavirus.jhu.edu/> (accessed 19 November 2021).
6. Ministry of Health, Labour and Welfare, Japan. The number of positive cases for Novel Coronavirus Disease. 2021. <https://www.mhlw.go.jp/content/10906000/000759133.pdf> (accessed 2 June 2021).
7. Shahnazi H, Ahmadi-Livani M, Pahlavanzadeh B, et al. Assessing preventive health behaviors from COVID-19: a cross sectional study with health belief model in Golestan Province, Northern of Iran. *Infect Dis Poverty* 2020;9(1):157. doi: 10.1186/s40249-020-00776-2
8. Chen X, Chen H. Differences in Preventive Behaviors of COVID-19 between Urban and Rural Residents: Lessons Learned from A Cross-Sectional Study in China. *Int J Environ Res Public Health* 2020;17(12) doi: 10.3390/ijerph17124437
9. Li S, Feng B, Liao W, et al. Internet Use, Risk Awareness, and demographic characteristics associated with engagement in preventive behaviors and testing: cross-sectional survey on COVID-19 in the United States. *J Med Internet Res* 2020;22(6):e19782. doi: 10.2196/19782
10. An L, Hawley S, Van Horn ML, et al. Development of a coronavirus social distance attitudes scale. *Patient Educ Couns* 2020 doi: 10.1016/j.pec.2020.11.027
11. Yihan X, Mark E, Tania L, et al. A small number of people account for a large amount of coronavirus risk. In: team Tbi, ed. United Kingdom: The Behavioural Insights Team, 2020. Available from: <https://www.microcovid.org> (accessed 20 November 2020).

VERSION 2 – REVIEW

REVIEWER	Chan, Ho Fai Queensland University of Technology
REVIEW RETURNED	13-Dec-2021

GENERAL COMMENTS	The authors have satisfactorily responded to all my questions and made the necessary changes to the manuscript. I therefore recommend publication of the study.
---

REVIEWER	Marin, Benjamin Brown University
REVIEW RETURNED	01-Dec-2021

GENERAL COMMENTS	Thank you for editing your manuscript. Your responses to my queries are appropriate and I appreciate the edits on the manuscript. The manuscript is much stronger now!
--